# A Novel Mutation in a Gene Causes Sclerosteosis in a Family of Mediterranean Origin

**DOI:** 10.3390/medicina58020202

**Published:** 2022-01-28

**Authors:** Aishah A. Ekhzaimy, Ebtihal Y. Alyusuf, Meshael Alswailem, Ali S. Alzahrani

**Affiliations:** 1Division of Endocrinology, Department of Internal Medicine, College of Medicine, King Saud University, Riyadh 11437, Saudi Arabia; ealyusuf82@gmail.com; 2Division of Molecular Endocrinology, Department of Molecular Oncology, King Faisal Specialist Hospital and Research Centre, Riyadh 11437, Saudi Arabia; malswailem10@kfshrc.edu.sa (M.A.); aliz@kfshrc.edu.sa (A.S.A.); 3Department of Medicine, King Faisal Specialist Hospital and Research Center, Riyadh 11437, Saudi Arabia

**Keywords:** *SOST*, sclerostin, sclerosteosis, Wnt signaling, bone, syndactyly

## Abstract

*Background and Objectives*: Sclerostin is an *SOST* gene product that inhibits osteoblast activity and prevents excessive bone formation by antagonizing the Wnt signaling pathway. Sclerosteosis has been linked to loss of function mutations in the *SOST* gene. It is a rare autosomal recessive disorder characterized by craniotubular hyperostosis and can lead to fatal cerebellar herniation. Our aim is to describe the clinical and radiological features and the new underlying *SOST* mutation in a patient with sclerosteosis. *Case:* A 25-year-old female who was referred to the endocrine clinic for suspected excess growth hormone. The patient complained of headaches, progressive blurred vision, hearing disturbances, increased size of feet, proptosis, and protrusion of the chin. She had normal antenatal history except for syndactyly. Images showed diffuse osseous thickening and high bone mineral density. Biochemical and hormonal tests were normal. Due to progressive compressive optic neuropathy, optic nerve fenestration with decompression hemicraniotomy was performed. Sclerosteosis was suspected due to the predominant craniotubular hyperostosis with syndactyly. Using peripheral leucocyte DNA, genomic sequencing of the *SOST* gene was performed. This identified a novel deletion homozygous mutation in the *SOST* gene (c.387delG, p.Asp131ThrfsTer116) which disrupts sclerostin function, causing sclerosteosis. *Conclusions*: Discovery of the molecular basis of sclerosteosis represents an important advance in the diagnosis and management of this fatal disease.

## 1. Introduction

Sclerosteosis is a rare autosomal recessive condition characterized by bone overgrowth and increased bone density. Sclerostin is secreted by osteocytes and plays a critical role inhibiting osteoblast activity and preventing excessive bone formation by antagonizing the Wnt signaling pathway [1]. To perform this function, sclerostin enhances its suppressive effect by binding to LDL receptor-related protein 5/6 (LRP5/6) [2]. Loss of function mutations of the *SOST* gene—a gene coding for sclerostin—are linked to sclerosteosis.

Cranial and tubular bones are the main bones affected by sclerosteosis, leading to gigantism, distortion of facial features, and cranial nerve entrapment. The presence of syndactyly is a key diagnostic feature. Sclerosteosis is a potentially lethal condition due to the associated increase in intracranial pressure and cerebellar herniation. The severity of the condition is inversely proportional to the abundance of sclerostin. No associated endocrine abnormalities have been noted or reported [3]. According to recent updates, around 96 patients have been reported worldwide [4]. The majority of affected individuals have been reported in South Africa [5,6]; however, a small number of individuals have been reported in other parts of the world [7,8,9,10,11,12]. Chromosomal region 17q12-q21 is responsible for two similar hyperostosis conditions, sclerosteosis and van Buchem disease (VBCH); however, they involve different loss of function mutations in the *SOST* gene (coding region in sclerosteosis and regulatory element in VBCH) [13,14]. VBCH is generally a milder sclerosing condition than sclerosteosis [15].

Due to its rarity, a knowledge gap exists with respect to the molecular basis of sclerosteosis, resulting in unsatisfactory diagnostic and therapeutic strategies. Discovery of the molecular basis of sclerosteosis represents an important advance in the diagnosis and management of this fatal disease. Here, we report a novel *SOST* gene mutation as the cause of sclerosteosis in a patient of Mediterranean origin (Syria) who is living in Saudi Arabia.

## 2. Case Presentation

A 25-year-old female was referred to our endocrine clinic for suspected growth hormone excess. The patient complained of headaches, progressive bilateral blurred vision and hearing disturbances, irregular menses, and generalized arthralgia that limited her mobility at the age of 23 years. Subsequently, she observed a progressive increase in the size of her feet and hands, change in facial contour, proptosis, and protrusion of the chin. She denied any history of fracture. She was the ninth of twelve siblings from consanguineous parents, originally from Syria. She had normal antenatal and neonatal history except for syndactyly. All family members were phenotypically normal except for a 33-year-old female sibling with similar physical appearance who had cranial decompression at the age of 20 years, a 20-year-old nephew with tall stature and syndactyly, and a 1-year-old nephew with nail dystrophy and possible metabolic bone disease. Those siblings were not available for evaluation. The father died at 60 years of age with acute respiratory failure (Figure 1).

The patient was tall with a height of 170.5 cm, weight of 105.4 kg, and body mass index (BMI) of 36.26 kg/m^2^. She had large facial contours, frontal bossing, prominent supraorbital ridges, proptosis of the eyes, prognathism, and misaligned teeth. Skeletal examination revealed a deformed left index finger with soft tissue syndactyly between the fingers of both hands and dysplastic nails. She had a normal visual acuity, visual field by confrontation, extraocular muscle movement, and bilateral papilledema.

A pituitary MRI was carried out initially due to suspicion of a pituitary adenoma and she was referred to the endocrine clinic to rule out acromegaly. The MRI revealed an enlarged sella turcica with a normal pituitary gland. Surprisingly, the MRI showed diffuse thickening of the osseous structures, including skull base, vault, and hard palate, with narrowing of skull base foramina, optic and internal auditory canals, secondary compression of cerebral parenchyma, and bilateral cerebellar tonsillar herniation (Figure 2A). Further imaging revealed a generalized increase in cortical thickness, vertebral end plate sclerosis, a deformed left index finger, and extremely increased bone mineral density with Z-score values of 12.8 at lumbar spine, 12.4 at left femoral neck, 12.5 at right femoral neck, and 7.1 at radius (Figure 2). Biochemical and endocrine tests revealed normal growth hormone (GH), insulin like growth factor-1 (IGF-1), thyroid stimulating hormone (TSH), prolactin, Short Synacthen test, follicular stimulating hormone (FSH), luteinizing hormone (LH), estradiol, calcium, phosphorus, parathyroid hormone (PTH), and alkaline phosphatase. Due to progressive worsening of vision caused by compressive optic neuropathy, optic nerve fenestration with decompressive hemicraniectomy was performed.

Sclerosteosis was suspected due to the predominant craniotubular hyperostosis with syndactyly. There is no curative therapy—only management aimed at relieving symptoms and preventing complications. The patient was started on calcitriol and prednisolone. Prednisolone could help in improving the bone pain and increasing bone resorption by stimulating the osteoclasts. Calcitriol could prevent the increase in PTH which might be induced by steroid. A multidisciplinary team approach was implemented by involving the ophthalmology, otolaryngology, neurosurgery, radiology, genetics, maxillofacial, and endocrinology teams in the management plan. We interviewed the family members and obtained a comprehensive history and conducted a full physical examination. After taking informed consent, we performed a whole exome sequencing of the patient followed by direct Sanger sequencing of several family members. The patient died suddenly with respiratory distress at the age of 27 years.

With informed consent and institutional review board approval (King Saud University Institutional Review Board (IRB), No. 21/01134/IRB, 26.12.2021 (22.05.1443)) we collected 5 cc blood from the patient, her other affected sister, mother, and three healthy brothers for DNA extraction and molecular testing. Genomic DNA was extracted from peripheral leucocytes using the Gentra Puregene blood kit (Catalog #158389, Qiagen, Valencia, CA, USA) according to the manufacturer’s instructions. Whole exome sequencing using the Ion Torrent platform (Thermo Fisher Scientific, Waltham, MA, USA) was carried out and the annotated sequencing revealed a *SOST* single nucleotide deletion mutation (c.387delG, p.Asp131ThrfsTer116). This was confirmed by polymerase chain reaction (PCR) and direct sequencing (Figure 3) of exon 2 of the *SOST* gene using the following primers: Forward, GCAGAGGACAGAAATGTGGG; reverse, CCACAACGTGTCCGAGTACAG. This mutation causes frameshift, abolishing the original stop codon and prolongation of the transcript by 34 amino acids (+34 AA). This likely leads to nonsense-mediated decay (NMD) of mRNA. This mutation was not reported in 1000 G (https://www.ncbi.nlm.nih.gov/variation/tools/1000genomes/ser, accessed on 20 December 2021) or ExAC databases (http://exac.broadinstitute.org, accessed on 20 December 2021) and is predicted to be disease-causing by MutationTaster (https://www.mutationtaster.org/, accessed on 20 December 2021) and PolyPhen2 (http://genetics.bwh.harvard.edu/pph2/, accessed on 20 December 2021) (Score 0.92). In addition to the prolonged transcript that is likely subject to NMD, the highly conserved negatively charged amino acid aspartic acid at codon 131 is changed to the polar uncharged amino acid threonine, with the rest of the transcript changed due to the frameshift.

## 3. Discussion

Bone is a metabolically active organ that undergoes continuous remodeling. Bone remodeling is maintained mainly by osteoclasts, osteoblasts, and osteocytes [16]. Imbalance between bone formation and bone resorption can lead to conditions with low or high bone mass. Sclerosteosis is an autosomal recessive condition characterized by an increase in the bone formation rate due to uncontrolled overactive osteoblast function [17]. Our patient had typical features of sclerosteosis in the form of progressive bone growth, tall stature, frontal bossing, prognathism, large mandible, dental malocclusion, progressive visual and hearing loss, and syndactyly. Hyperostosis of the calvarium increased the intracranial pressure leading to cerebellar herniation which was managed by craniotomy. In addition to her clinical presentation, the patient’s radiographs showed increased bone density suggesting bone sclerosing dysplasia. The skeletal manifestations of sclerosteosis are due to endosteal hyperostosis leading to progressive generalized osteosclerosis [4].

In the present study, we describe a new mutation in the *SOST* gene in two female siblings from consanguineous Syrian parents. This mutation is expected to result in a loss of function of sclerostin. The presence of heterozygous mutation of the gene in some siblings and the mother is consistent with the autosomal recessive nature of this condition, where the presence of two mutant alleles could only possibly result in development of the disease phenotype, since a functional test on the mutant protein has not been conducted. Each sibling of an affected individual has a 25% chance of inheriting the two alleles and being affected, a 25% chance of being unaffected and not a carrier, and a 50% chance of being an asymptomatic carrier of one allele. *SOST* is a two-exon gene and encodes sclerostin, a 190-amino acid glycoprotein of the DAN family [18]. Sclerostin has been shown to have an antagonistic effect on the LDL receptor-related protein (LRP) 5/6-mediated canonical Wnt signaling pathway—a signaling cascade of the osteoblastic bone formation—by direct binding to LRP5/6 [19]. Therefore, the absence of sclerostin in the bones of patients with sclerosteosis may result in hyperactivation of Wnt signaling, leading to bone overgrowth. Understanding the role of sclerostin in bone metabolism and the genetic mutation behind rare diseases such as sclerosteosis have helped in the development of new therapeutics for the treatment of common diseases such as osteoporosis [20]. Studies have shown that targeting sclerostin with romozosumab, a sclerostin monoclonal antibody, is an effective strategy that increases bone mineral density and reduces fracture risk in postmenopausal women and patients with multiple myeloma with osteoporosis [21,22].

Mutations in the *SOST* gene have been associated with inherited high bone mass conditions. The clinical spectrum of these conditions ranges from severe craniodiaphyseal dysplasia to non-pathological high bone mass [23,24,25,26]. The most severe form is craniodiaphyseal dysplasia, an extremely rare autosomal dominant condition. Sclerosteosis is another severe but autosomal recessive condition. In contrast, VBCH is a mild sclerosing disease. Several mutations have been reported in patients with sclerosteosis in different countries around the world. These mutations result in loss of function of sclerostin and include missense, nonsense, frameshift, and splice site mutations (Table 1).

In patients with osteopetrosis, a condition similar to sclerosteosis, the sclerotic bone is primarily found in epiphysis, metaphysis, and diaphysis of long bones which are liable to fracture [29]. In our patient, despite diffuse sclerosis, the patient had no fractures. Sclerosteosis must be differentiated from VBCH, another craniotubular hyperostosis condition that is caused by the deletion of a *SOST*-specific regulatory element [14]. Sclerosteosis is predominantly found in people from South Africa and affected patients are usually tall and have an early finding of syndactyly, which can be detected during the neonatal period. The thickening of the calvarium in sclerosteosis can lead to a lethal increase in intracranial pressure. Tall stature and syndactyly are the main distinctive features between sclerosteosis and VBCH. VBCH is generally milder than sclerosteosis with no associated tall stature nor syndactyly. Patients with VBCH have normal life spans and almost always originate from the Netherlands [4,5,15]. Another differential diagnosis is acromegaly, a more commonly thought of diagnosis, due to the facial features; however, it was excluded in our patient by normal IGF-1 and by the radiological features.

Current treatment for sclerosteosis is limited. A recent study investigating the effectiveness of sclerostin replacement in a mouse model of sclerosteosis found that sclerostin replacement in mice partially corrected the high bone mass phenotype of affected mice. However, its modest efficacy in the presence of excessive bone formation in sclerosteosis suggests that it may not be an optimal therapy [30].

## 4. Conclusions

We describe a novel mutation in the *SOST* gene in a patient with sclerosteosis, that has not been previously described. It is evident that sclerosteosis is a severe fatal disorder which places a considerable burden upon affected individuals. Understanding the underlying molecular mechanisms is not only of great benefit to confirm the diagnosis but may pave the way for future therapeutic development for this progressive and often fatal condition.

## Figures and Tables

**Figure 1 medicina-58-00202-f001:**
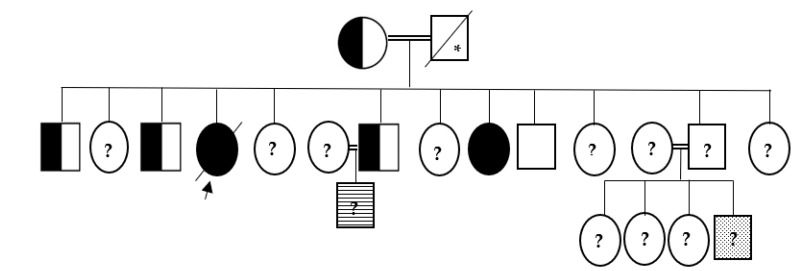
Family pedigree. Filled black symbols indicate affected family members with sclerosteosis (SOST ex.2a: c.387delG, p.Asp131ThrfsTer116) homozygote. Half-filled symbols indicate heterozygote for the mutation. The dotted symbol indicates having syndactyly and the lined one indicates nail dystrophy with query bone metabolic bone disease. The question mark symbols indicate unknown mutation status (not evaluated). The star symbol indicates inferred mutation carrier (heterozygote or homozygote). The proband is indicated with the arrow.

**Figure 2 medicina-58-00202-f002:**
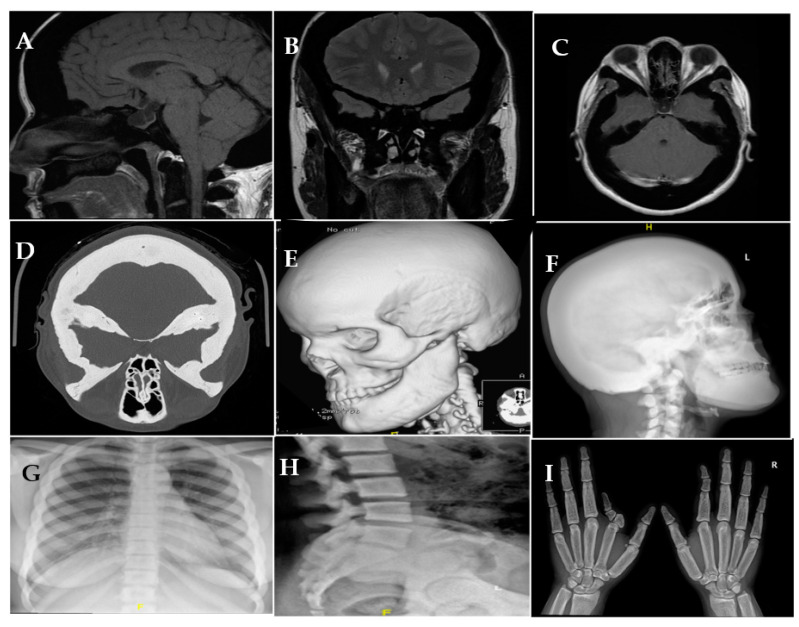
(**A**) Magnetic resonance imaging (MRI) T1 sagittal view: Enlarged sella turcica filled with CSF. Normal appearing pituitary gland at base of sella. Diffuse thickening of the osseous structures. tonsillar ectopia, cerebellar tonsillar herniation 8 mm below foramen magnum. (**B**) MRI: Severe narrowing of the skull base foramina including optic canals with optic nerve compression. (**C**) MRI T1 axial view: Loss of bone marrow fat and increased bone thickness. (**D**) Computed tomography (CT) bone window: Severe diffuse osseous thickening. (**E**) 3-D CT scan of the brain: Severe diffuse thickening of the osseus structures of the skull, facial bones. (**F**–**I**) X-ray bone survey: Generalized increase in bone density with diffuse cortical thickening and deformed left second metacarpal head.

**Figure 3 medicina-58-00202-f003:**
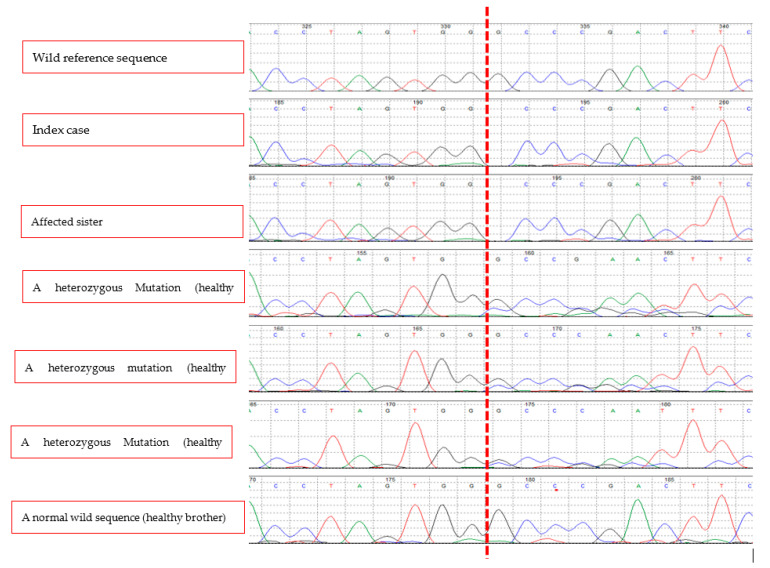
A biallelic deletion mutation (c.387delG, p.Asp131ThrsTer116) in exon 2 of *SOST* gene of the index case. Her affected sister has the same mutation while two of her unaffected brothers and her mother have the same mutation in a monoallelic form. One healthy brother has a normal sequence. The vertical red dashed line indicates the location of the deleted nucleotide (G) in homozygous form in the two affected sisters, in a homozygous form in the unaffected carriers, and a normal sequence in the healthy brother.

**Table 1 medicina-58-00202-t001:** The reported mutations in *SOST* gene leading to sclerosteosis.

Reference	Parental Consanguinity	Origin	Syndactyly	DNA Nucleotide Change	Predicted Protein Change	Mutation Effect
Whyte et al., 2018 [10]	Yes	India	Yes	*c.129C > G*	p.Try43X	Nonsense
He et al., 2016 [12]	Yes	China	Yes	*(c.444_445TC > AA) in exon 2*	p.Cys148Ter	Premature stop codon
Fayez et al., 2015 [8]	Yes	Egypt	Yes	*c.87_88insC in exon 1*	p.Lys30GInfsTer3	Frame shift
Yagi et al., 2015 [27]	Yes	Turkey	Yes	*c.371G > A (p.W124*) in exon 2*	pTrp124Ter	Nonsense
Belkhribchia et al., 2014 [9]	No	Morocco	Yes	*c.79C > T (p.Gln27*) in exon 1*	p.Gln27Ter	Nonsense
Bhadada et al., 2013 [11]	No	India	Yes	*c.296_297insC in exon 2*	(p.Val100fsX128)	Frame shift
Piters et al., 2010 [1]	Yes	Turkey	No	*(c.499T > C) in exon 2*	p.Cys167Arg	Missense
Kim et al., 2008 [28]	No	Brazil	Yes	*c.372G > A (Trp124X) exon 2*	p.Trp124Ter	Nonsense
Balemans et al., 2001 [24]	NK	USA	NK	*c.376C > T*	p.Arg126Ter	Nonsense
Balemans et al., 2001 [24]	Yes	Brazil	NK	*c.372G > A (Trp124X) in exon 2*	p.Trp124Ter	Nonsense
Brunkow at al., 2001 [13]	6: Yes, 2: NK21: No	South Africa	NK	*c.69C > T*	p.Gln24Ter	Nonsense
Tacconi et al., 1998 [6]	NK	Senegal	Yes	*IVS1 + 3A > T*	NK	Splice site

NK: Not known.

## Data Availability

All the data are available from the corresponding author upon reasonable request.

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
