# Peer review of "A Novel Mutation in a Gene Causes Sclerosteosis in a Family of Mediterranean Origin"

_medicina, 2022, doi:10.3390/medicina58020202_

Round 1

Reviewer 1 Report

The identification of a new mutation in the SOST gene would add knowledge to the field of the study of sclerosteosis, a rare autosomal recessive inherited bone disorder.  While functional test on the mutant protein has not been conducted, it should be cautious in interpreting the effect of the mutation on the symptoms.  It should be indicated that the “loss of function” is a possibility only.  A few more questions below needs to be addressed:

  1. The mother is a heterozygote and the father is not a carrier of the mutation as shown in Figure 1. In this case how can two of their daughters contain the mutation in both of the two alleles and become affected patients with sclerosteosis as this is an autosomal recessive mutation?  If the father’s genotype is an inferred mutation carrier (heterozygote of homozygote), it should be indicated in the pedigree.
  2. Biopsy was obtained from six members only according to the description (Lines 133-135. Therefore, the genotypes of the other siblings/members should be unknown.  However, their genotypes are presented as normal (non-carrier) in the pedigree (Figure 1).  This is not objective.
  3. Figure 3: The DNA sequence from five of the six member analysed are presented. Why the sequence of the mother is not?  It should be presented as well.

Author Response

Manuscript ID: medicina-1543083: A Novel Mutation in the Gene Causes Sclerosteosis in a Family of Mediterranean Origin

Authors’ Response to Reviewers

We are grateful for the reviewers for their insightful comments and valuable feedback on our manuscript. We have endeavored to address all concerns and feel that the manuscript has been strengthened as a result. We have carefully revised the text in response to these comments. Specific revisions are detailed below and the boldface text indicates revisions in the manuscript. We have also attached the revised manuscript with track changes. Here is a point-by-point response to the reviewers’ comments and concerns

Reviewer 1:

Comments to Authors:

“The identification of a new mutation in the SOST gene would add knowledge to the field of the study of sclerosteosis, a rare autosomal recessive inherited bone disorder.  While functional test on the mutant protein has not been conducted, it should be cautious in interpreting the effect of the mutation on the symptoms.  It should be indicated that the “loss of function” is a possibility only”

Response to reviewer:

We appreciate the reviewer’s comment. We have now clarified this in the “Discussion” section (lines 208-211) as follows:

The presence of heterozygous mutation of the gene in some siblings and the mother is consistent with the autosomal recessive nature of this condition, where the presence of two mutant alleles could only possibly result in development of the disease phenotype, since functional test on the mutant protein has not been conducted.

Comments to Authors:

  1. “The mother is a heterozygote and the father is not a carrier of the mutation as shown in Figure 1. In this case how can two of their daughters contain the mutation in both of the two alleles and become affected patients with sclerosteosis as this is an autosomal recessive mutation?  If the father’s genotype is an inferred mutation carrier (heterozygote of homozygote), it should be indicated in the pedigree.”

Response to reviewer:

Thank you for pointing this out. We have edited this to the pedigree in figure 1 as suggested by the reviewer. We have also edited the figure’s legend accordingly in the “Case Presentation” section (lines 81-86). 

Figure 1. Family pedigree. Filled black symbols indicate affected family members with sclerosteosis (SOST ex.2a: c.387delG, p.Asp131ThrfsTer116) homozygote. Half-filled symbols indicate heterozygote for the mutation. The dotted symbol indicates having syndactyly and the lined one indicates nail dystrophy with query bone metabolic bone disease. The question mark symbols indicate unknown mutation status (not evaluated). The star symbol indicates inferred mutation carrier (heterozygote or homozygote). The proband is indicated with the arrow.

Comments to Authors:

  1. “Biopsy was obtained from six members only according to the description (Lines 133-135. Therefore, the genotypes of the other siblings/members should be unknown.  However, their genotypes are presented as normal (non-carrier) in the pedigree (Figure 1).  This is not objective.”

Response to reviewer:

Thank you for pointing this out. We have edited this to the pedigree in figure 1 as suggested by the reviewer. We have also edited the figure’s legend accordingly in the “Case Presentation” section (lines 81-86). 

Figure 1. Family pedigree. Filled black symbols indicate affected family members with sclerosteosis (SOST ex.2a: c.387delG, p.Asp131ThrfsTer116) homozygote. Half-filled symbols indicate heterozygote for the mutation. The dotted symbol indicates having syndactyly and the lined one indicates nail dystrophy with query bone metabolic bone disease. The question mark symbols indicate unknown mutation status (not evaluated). The star symbol indicates inferred mutation carrier (heterozygote or homozygote). The proband is indicated with the arrow.

Comments to Authors:

  1. “Figure 3: The DNA sequence from five of the six member analysed are presented. Why the sequence of the mother is not?  It should be presented as well.”

Response to reviewer:

Thank you for the comment. We have now included all family members with available DNA sequence in the new figure 3.

Reviewer 2 Report

Dear Authors, I just complete the revision of your manuscript. 

I think that studying rare diseases is always amazing because of all the news could be applied to general disease such as osteoporosis. Sclerosteosis has been used to improve therapies for osteoporosis.

I have some minor suggetion to improve your paper. 

You have some more info on the affected sister? It could be interesting to comment 2 patients with the same deletion.

pag 4. "the patietn started calcitriol and prednisolone to suppress the osteoclasts" You can clarify this point?

Conflict of interest: you may state correctly if yes/no interest are present?

Moreover I suggest to insert the citation of this 2 recent review/article about bone diseases/sclerosteosis. The first about the importance of rare disease to improve the novelty of general disease. The second on sclerosteosis treatment in vitro and in vivo. 

_Rossi, Battafarano, De Martino, Minisola, Del Fattore. Looking for new anabolic treatment from rare diseases of bone formation. J Endocrinol 2020 Dec 1 doi:10.1530/JOE-20-0285

_Dreyer, Shah, Doyle, Greenslade, Penney, Creeke, Kotian, Ke, Naidoo, Holdsworth. Recombinant sclerostin inhibits bone formation in vitro and in mouse model of sclerosteosis. J Orthop Translat 2021 Jun 21. doi: 10,1016/j.jot.2021.05.005

Author Response

Manuscript ID: medicina-1543083: A Novel Mutation in the Gene Causes Sclerosteosis in a Family of Mediterranean Origin

Authors’ Response to Reviewers

We are grateful for the reviewers for their insightful comments and valuable feedback on our manuscript. We have endeavored to address all concerns and feel that the manuscript has been strengthened as a result. We have carefully revised the text in response to these comments. Specific revisions are detailed below and the boldface text indicates revisions in the manuscript. We have also attached the revised manuscript with track changes. Here is a point-by-point response to the reviewers’ comments and concerns

Reviewer 2:

Comments to Authors:

  1. “Dear Authors, I just complete the revision of your manuscript.I think that studying rare diseases is always amazing because of all the news could be applied to general disease such as osteoporosis. Sclerosteosis has been used to improve therapies for osteoporosis. I have some minor suggetion to improve your paper. “

Response to reviewer:

We appreciate the reviewer’s comment.

Comments to Authors:

  1. “You have some more info on the affected sister? It could be interesting to comment 2 patients with the same deletion.”

Response to reviewer:

Thank you for the comment. Unfortunately, we don’t have any more information about her sister

Comments to Authors:

  1. pag 4. "the patietn started calcitriol and prednisolone to suppress the osteoclasts" You can clarify this point?

Response to reviewer:

Thank you for the comment. In response to the reviewer, we have clarified this point in the “Case Presentation” section (lines 128-130)

“Prednisolone could help in improving the bone pain and increasing bone resorption by stimulating the osteoclasts. Calcitriol could prevent the increase in PTH which might be induced by steroid.

Comments to Authors:

  1. “Conflict of interest: you may state correctly if yes/no interest are present?”

Response to reviewer:

Thank you for the comment. We have corrected this statement (lines 278-279).

Conflicts of Interest: None.

Comments to Authors:

  1. “Moreover I suggest to insert the citation of this 2 recent review/article about bone diseases/sclerosteosis. The first about the importance of rare disease to improve the novelty of general disease. The second on sclerosteosis treatment in vitro and in vivo. 

_Rossi, Battafarano, De Martino, Minisola, Del Fattore. Looking for new anabolic treatment from rare diseases of bone formation. J Endocrinol 2020 Dec 1 doi:10.1530/JOE-20-0285

_Dreyer, Shah, Doyle, Greenslade, Penney, Creeke, Kotian, Ke, Naidoo, Holdsworth. Recombinant sclerostin inhibits bone formation in vitro and in mouse model of sclerosteosis. J Orthop Translat 2021 Jun 21. doi: 10,1016/j.jot.2021.05.005”

Response to reviewer:

Thank you for the comment. In response to the reviewer, we have made some edits in the “Discussion section” (lines 220-222) and (lines 254-258) and we included these two recent additional references (ref# 20 and 30). We have also updated the order of the references section accordingly. 

(lines 220-222)

“Understanding the role of sclerostin in bone metabolism and the genetic mutation behind rare diseases such as sclerosteosis have helped in the development of new therapeutics for the treatment of common diseases like osteoporosis.”

(lines 254-258)

“Current treatment for sclerosteosis is limited. A recent study investigated the effectiveness of sclerostin replacement in a mouse model of sclerosteosis founds that sclerostin replacement in mice partially corrected the high bone mass phenotype of affected mice. However, its modest efficacy in presence of excessive bone formation in sclerosteosis suggests that it may not be an optimal therapy”
